



**The impact of spatiotemporal structure of rainfall on flood**
**frequency over a small urban watershed: an approach coupling**
**stochastic storm transposition and hydrologic modeling**
Zhengzheng Zhou[1], James A. Smith[2], Mary Lynn Baeck[2], Daniel B. Wright[3], Brianne K. Smith[4],
Shuguang Liu[1]
[1]Department of Hydraulic Engineering, Tongji University, Shanghai, China.
[2]Department of Civil and Environmental Engineering, Princeton University, USA.
[3]Department of Civil and Environmental Engineering, University of Wisconsin-Madison, USA.
[4]Department of Earth and Environmental Sciences, City University of New York-Brooklyn College, USA.
*Corresponding to*: Zhengzheng Zhou (zhouzz@tongji.edu.cn); Shuguang Liu (liusgliu@tongji.edu.cn)
**Abstract.** The role of rainfall space-time structure, as well as its complex interactions with land surface properties, in
flood response remains an open research issue. This study contributes to this understanding, specifically in small (<15 km$^2$)
urban watersheds. Using a flood frequency analysis framework that combines stochastic storm transposition-based rainfall
scenarios with the physically-based distributed GSSHA model, we examine the role of rainfall spatial and temporal
variability in flood frequency across drainage scales in the highly-urbanized Dead Run watershed (14.3 km$^2$) outside of
Baltimore, Maryland, USA. The results show the complexities of flood response within several subwatersheds for both
short (<50 years) and long (>100 years) rainfall return periods. The impact of impervious area on flood response decreases
with increasing rainfall return period. For extreme storms, the maximum discharge is closely linked to the spatial structure
of rainfall, especially storm core spatial coverage. The spatial heterogeneity of rainfall increases flood peak magnitudes by
50% on average at the watershed outlet and its subwatersheds for both small and large return periods. The results imply
that commonly-made assumption of spatially uniform rainfall in urban flood frequency modeling is problematic even for
relatively small basin scales.
**1. Introduction**
Rainfall spatiotemporal structure plays an important role in flood generation in urban watersheds (Ogden *et al.*, 1995;
Saghafian *et al.*, 1995; Smith *et al.*, 2005b; Emmanuel *et al.*, 2012; Nikolopoulos *et al.*, 2014). Spatial heterogeneities in
land use and land cover complicate the translation of rainfall spatiotemporal distribution into flood responses (Galster *et*
*al.*, 2006; Morin *et al.*, 2006; Ntelekos *et al.*, 2008; Ogden *et al.*, 2011), especially for small catchments (Faurès *et al.*, 1995;
Smith *et al.*, 2005a; Zhou *et al.*, 2017). Due to the varying nature of rainfall and complexities of urban characteristics, the
influence of rainfall spatial-temporal structure on flood frequency analysis in urban areas remains an open research issue.





Many studies have examined the interaction between rainfall variability and flood response. By necessity, early studies
tended to explore rainfall variability using rain gages, which were the main source of rainfall measurements until relatively
recently. The accuracy of flood simulations using spatially-detailed rainfall scenarios has been examined (Dawdy and
Bergmann, 1969; Schilling, 1991), along with the sensitivity of hydrologic response to rainfall gage network density
(Faurès *et al.*, 1995; Arnaud *et al.*, 2002; Younger *et al.*, 2009; Notaro *et al.*, 2013). Beven and Hornberger (1982) argued
that the spatial variability affects the response time more than the peak magnitude, whereas Wilson *et al.* (1979) found the
reverse. These studies were limited, however, by the general sparsity of rain gages, which may not adequately capture the
spatial distribution of rainfall. Following the advent of rainfall measurement using weather radar (Fulton *et al.*, 1998;
Krajewski and Smith, 2002), many studies have highlighted the use of high-resolution rainfall data in assessing rainfall
variability over various range of spatial and temporal scales (Berne *et al.*, 2004; Gebremichael and Krajewski, 2004;
Moreau *et al.*, 2009; Emmanuel *et al.*, 2012) and how their use could improve runoff estimation (Morin *et al.*, 2006; Smith
*et al.*, 2007; Schellart *et al.*, 2012; Wright *et al.*, 2014b; Bruni *et al.*, 2015; Rafieeinasab *et al.*, 2015; Gourley *et al.*, 2017).
There are conflicting findings on the relative importance of rainfall temporal and spatial characteristics. Ochoa-Rodriguez
*et al.* (2015) and Yang *et al.* (2016), for example, found that "coarsening" temporal resolution has a stronger impact than
coarsening spatial resolution, especially for small watersheds. Similar results were found in the study of Paschalis *et al.*
(2014) in a 477 km$^2$ catchment in Switzerland. Adams *et al.* (2012) found the space-time averaging effects of routing
through the catchment noticeably remove the impact of spatially variable rainfall at a 150-km$^2$ catchment scale. Bruni *et*
*al.* (2015), in contrast, found a higher sensitivity of modeled flow peaks to spatial resolution rather than the temporal
resolution. Peleg *et al.* (2017) showed an increasing contribution of the spatial variability of rainfall to the variability of
flow discharge with longer return periods. Cristiano *et al.* (2018); Cristiano *et al.* (2019) found the spatial aggregation of
rainfall data can have a strong effect on hydrological responses. Zhu *et al.* (2018) examined the influence of rainfall
variability on flood frequency analysis and addressed the impact of antecedent moisture in flood generation for basin scales
ranging from 16 km$^2$ up to 4,400 km$^2$. Using observational data, Zhou *et al.* (2017) showed that the impact of antecedent
moisture is low in a highly-urbanized catchment.
Previous studies have demonstrated the sensitivity of hydrological response to rainfall variability in both space and time
(Smith *et al.*, 2012; Ochoa-Rodriguez *et al.*, 2015; Rafieeinasab *et al.*, 2015). The relationship between rainfall and flood
are scale-dependent, varying with rainfall patterns, basin characteristics, and runoff generation processes. However, there
is still no clear answer on the relative importance of temporal and spatial features of rainfall on flood responses (Cristiano
*et al.*, 2017). Moreover, studies focusing on small (< 15 km$^2$) urbanized basins are relatively few (Peleg *et al.*, 2017) and
the issues remain poorly understood.



Stochastic Storm Transposition (SST) was developed as a physically-based stochastic rainfall generator for rainfall
frequency analysis. Previous studies show that SST with relatively short-term records (10 or more years) of high-resolution
radar rainfall field can produce reasonable rainfall scenarios with spatial-temporal structure, which cannot be provided by
conventional methods (Wright *et al.*, 2013; Wright *et al.*, 2017; Zhou *et al.*, 2019). Coupled with hydrological models, the
SST-based framework can be used for multiscale rainfall frequency analysis and flood frequency analysis that accounts for
rainfall variability and surface characteristics (Wright *et al.*, 2014a; Perez *et al.*, 2019; Yu *et al.*, 2019; Wright *et al.*, 2020).
This study contributes to the interaction between rainfall variability and flood response over small-scale urbanized
watersheds (<15 km$^2$) for a short-duration rainfall and quick hydrologic response setting. We build on the SST-based rainfall
study of Zhou *et al.* (2019) using the physically-based hydrological model implementation introduced by Smith *et al.* (2015)
in the Dead Run watershed outside of Baltimore, Maryland, USA by addressing the following questions: (1) How does
flood frequency in small urban watersheds vary with diverse space-time rainfall structure and rainfall magnitude? (2)
Among the space-time feature of rainfall, what are the dominant features that control flood peak distribution in small urban
watersheds?
Using a framework that combines high-resolution realistic SST- and radar-based rainfall scenarios with model-based flood
frequency analysis, we characterize the spatial and temporal features of rainfall events under different return periods and
examine their roles in determining flood frequency in small urban watersheds. The paper is organized as follows: in Section
2, we introduce the study region and describe the SST-based methodology, GSSHA model, and the metrics used to
characterize rainfall and flood response. In Section 3, we present model validation and analyses of flood frequency
distributions and rainfall-flood relationships. A summary and conclusions are presented in Section 4.
**2. Data and method**
**2.1 Study region and data**
The study focuses on the highly-urbanized 14.3 km$^2$ Dead Run (DR) watershed located west of Baltimore, Maryland, USA
(Fig. 1). DR is a tributary to the Gwynns Falls watershed, which is the principal study catchment of the Baltimore
Ecosystem Study (BES) (Pickett and Cadenasso, 2006). The basin has an impervious fraction of approximately 52.3%
(Table 1). The watershed has a dense network of six stream gauges with drainage areas ranging from 1.2 to 14.3 km$^2$ (Fig.
1; Table 1). The subwatersheds are developed after the implementation of the Maryland Stormwater Management Act of
1982 (Maryland, 1982) with many detention infrastructures such as small local ponds. The wealth of data for Dead Run
provides exceptional resources to examine rainfall and hydrologic response (Beighley and Moglen, 2002; Nelson *et al.*,
2006; Meierdiercks *et al.*, 2010; Smith *et al.*, 2015). For example, Meierdiercks *et al.* (2010) analyzed the impact of storm



drains and detention basins on a single storm event in DR, while Ogden *et al.* (2011) used the Gridded Surface Subsurface
Hydrologic Analysis (GSSHA) model to analyze the effects of storm drains, impervious area, and drainage density on
hydrologic response. Smith *et al.* (2015) created a DR model using GSSHA to examine the effects of storage and runoff
generation processes through analyses of a large number of storm events.

High resolution (15-min temporal resolution, 1-km$^2$ spatial resolution) radar rainfall fields for the 2000-2015 period were
derived from volume scan reflectivity fields from the Sterling, Virginia WSR-88D (Weather Surveillance Radar-1988
Doppler) radar. The Hydro-NEXRAD algorithms (Krajewski *et al.*, 2011; Seo *et al.*, 2011) which have been used in rainfall
and hydrological studies (Smith *et al.*, 2007; Lin *et al.*, 2010; Smith *et al.*, 2013; Wright *et al.*, 2014b; Zhou *et al.*, 2017)
are used to estimate rainfall from reflectivity fields. A network of 54 rain gauges in and around Baltimore City is used for
mean field bias correction of the radar rainfall. The reader is directed to Zhou *et al.* (2019) and references therein for further
details on the rainfall data and bias correction methods.
Instantaneous discharge data with a resolution of five minutes from the U.S. Geological Survey (USGS) were used for each
of the six gaged watersheds. Streamflow observations for the outlet station at Franklintown extend back to 1960, while the
DR1 – DR5 stations have records beginning in 2008.

**2.2 GSSHA Hydrological Model**

The distributed physics-based GSSHA model is used to simulate multi-scale flood response. GSSHA is a two-dimensional,
distributed-parameter raster-based (i.e. square computational cell-based) hydrologic modeling system. It uses explicit finite
difference and finite volume methods in two dimensions on a structured grid to simulate overland flow and in one dimension
to simulate channel flow (Downer and Ogden, 2004; 2006). Previous studies of the GSSHA model show that the model
with fine grid resolution can produce adequate simulations of flood response, especially when driven by high-resolution
radar rainfall fields (Sharif *et al.*, 2010; Sharif *et al.*, 2013; Wright *et al.*, 2014a; Cristiano *et al.*, 2019).
In this study, we use the Dead Run model created by Smith *et al.* (2015). A brief description of the model is provided here;
see Smith *et al.* (2015) for more details. The delineation of the watershed and channel network was based on a 30-m USGS
digital elevation model (Gesch *et al.*, 2002). Channel flow overland flow was set with different Manning's roughness
coefficients. Additional stream channels were added based on the Baltimore County hydrography Geographic Information
Systems   (GIS) map. Stream cross sections were extracted from a 1-m resolution topography data set for Dead Run
developed from lidar. Storm sewers in DR-2 and DR-5 were added using the Baltimore County Stormwater Management
GIS map and digitized storm sewer maps. The semicircle's diameter was set to the pipe diameter. Detention basins were



represented within the channel with cross sections extracted from the 1-m lidar topographic data.
Several aspects of the model were modified from those used in Smith *et al.* (2015), primarily to improve computational
speed. Infiltration is calculated using Richards' equation (RE) in Smith *et al.* (2015), while this study uses the three-layer
Green-Ampt (GA) scheme. A uniform Manning's roughness coefficient of 0.01 is set for all the stream channels for model
simplification. Initial soil moisture is approximated to be one third of field capacity for each storm event.
**2.3 SST procedure**
The rainfall scenarios in this study are developed using RainyDay, an open-source SST software package (Wright *et al.*,
2017). The steps used are briefly summarized here; the reader is directed to Zhou *et al.* (2019) and references therein for
further details.
The first step is to identify a geospatial "transposition domain" that contains the watershed of interest. In this study, we use
a square 7,000 km² transposition domain centered on the DR watershed. (Zhou *et al.*, 2019) presented a detailed
examination of heterogeneity in extreme rainfall over the transposition domain using a variety of metrics, including storm
counts, mean storm depths and intensities, convective activity indicated by lightning observations, and analysis of spatial
and temporal rainfall structure.
The second step is to identify the largest *m* storms within the domain at the *t*-hr time scale. This collection of storms is
referred to as a storm catalog. The storms are selected with respect to the size, shape and orientation of the DR watershed.
We henceforth refer to these as "DR-shaped storms." The *m* DR-shaped storms are selected from an *n*-year rainfall record,
such that an average of $\lambda=m/n$ storms per record year are included in the storm catalog. In this study, we chose $m = 200$
storms over the 16-year radar record.
The third step is to randomly sample a subset of *k* storms from the storm catalog, where *k* refers to a stochastic number of
storms per year. The *k* is assumed to follow a Poisson-distributed number of storm occurrences with rate parameter $\lambda=m/n$
storms per year. All rainfall fields associated with a storm are transposed by an east-west distance $\Delta x$ and a north-south
distance $\Delta y$, where $\Delta x$ and $\Delta y$ are drawn from distributions $D_X(x)$ and $D_Y(y)$ which are bounded by the limits of the
transposition domain. Based on the spatial heterogeneity analysis of extreme rainfall in the domain, distributions $D_X(x)$ and
$D_Y(y)$ can be set as uniform or non-uniform. In Zhou *et al.* (2019) and this study, since the assumption of regional
homogeneity cannot be relaxed, we used the non-uniform distribution. A two-dimensional probability density function
(PDF) of spatial storm occurrence (Wright *et al.*, 2017) and an intensity factor that rescales the rainfall magnitude (Zhou
*et al.*, 2019) are used as the basis for non-uniform spatial transposition (Fig. A1 in Appendix A). This step can be understood
as generating a "synthetic year" of extreme rainfall events over the domain based on resampling and transposing



observations. For each of the $k$ transposed storms, compute the $t$-hr basin-average rainfall depth over the watershed. Among
the $k$ rainfall depths, the maximum depth is retained as a synthetic $t$-hr annual rainfall maximum for the watershed, while
the transposed rainfall fields are saved for use as inputs to a GSSHA model simulation.
The fourth step simply repeats Step 3 many times to recreate multiple years of synthetic $t$-hour "annual" rainfall maxima
and associated transposed rainfall fields for the watershed. In this study, these steps are repeated 1,000 times and the ordered
"annual" maxima are used to generate rainfall return period estimates up to 500 years. 1,000 such realizations of 500-yr
series are generated, and the median value of 1,000 realizations are used to generate estimates for return periods up to 500
years.
**2.4 Characteristics of rainfall and hydrologic response**
**2.4.1 Spatio-temporal characteristics of rainfall**
Rainfall statistics were computed for each event, based on radar rainfall data at 15-min, 1-km$^2$ resolution, to characterize
the spatial and temporal variability of rainfall (following Smith *et al.* (2002); Smith *et al.* (2005b); see also Zoccatelli *et al.*
(2011) and Emmanuel *et al.* (2015)). Basin-average rainfall rate at time $t$ during the storm is given by:
$$M(t) = \int_0^T R(t,x)dx \tag{1}$$
where $R(t,x)$ is the rain rate at radar grid $x$ at time $t$, and $T$ is the time period of rainfall event. Peak basin-average rainfall
rate is denoted:
$$M_{max} = \max\{M(t); t \in [0,T]\} \tag{2}$$
and storm total rainfall depth is:
$$R_{sum} = \sum_0^T M(t) \tag{3}$$
To characterize the spatial properties of rainfall, several quantities are computed. Fractional coverage of storm core at $t$ is
given by:
$$Z(t) = \frac{1}{A}\int_A I_{(R(t,x))}dx \tag{4}$$
where $I_{(R(t,x))}$ is the indicator function and equals 1 when $R(t,x) > 25\ mm/h$ or 0 otherwise.
Rainfall location is given by:
$$L(t) = \int_A \omega(t,x)d(x)dx \tag{5}$$





where $\omega(t,x) = \frac{R(t,x)}{\int_A R(t,x)dx}$, $d(x)$ is the linear distance from point x to the outlet.    The rainfall-weighted flow distance is:
$$\text{RWD(t)} = \int_A \omega(t,x)d_f(x)dx \tag{6}$$
where distance function $d_f(x)$ is the flow distance between point x and the outlet. It is calculated as the sum of the
overland flow distance from $x$ to the nearest channel and the distance along the channel to the basin outlet. The flow distance
$d_f(x)$ is normalized by the maximum flow distance, ranging from 0 to 1. RWD with values close to 0 indicates that rainfall
is distributed near the basin outlet; with values close to 1 indicates rainfall concentrated at the far periphery of the basin.
For a uniformly distributed rainfall, the mean RWD is:
$$\overline{\text{RWD}} = \int_A d_f(x)dx \tag{7}$$
The dispersion of RWD:
$$\text{S(t)} = \frac{1}{\bar{s}}\int_A \omega(t,x)[d_f(x)-\bar{d}]^2dx \tag{8}$$
where $\bar{s} = \int_A [d_f(x)-\bar{d}]^2dx$, S is a spatial indicator with values < 1 indicates that rainfall is a unimodal distribution; S
with values >1 indicates that rainfall is a multimodal distribution.
**2.4.2 Spatiotemporal characteristics of hydrologic response**
Flood peak ($Q_{peak}$, mm³/s), total runoff ($Q_{sum}$, mm), and lag time ($T_{lag}$, min) are defined as:
$$Q_{peak} = \max\{ Q(t); t \in T_d \} \tag{9}$$
$$Q_{sum} = \sum_0^{T_d} Q(t) \tag{10}$$
$$T_{lag} = T_{Fpeak} - T_{Rpeak} \tag{11}$$
Respectively, where $Q(t)$ is the flow discharge at time $t$; $T_d$ is the duration of hydrological response, which is from the
start of rainfall event to the time when $f(t)<0.05*Q_{peak}$.
**3. Results and Discussion**
**3.1 Model validation**
We validated the Dead Run GSSHA model through analyses of the 21 largest warm season (April-September) flood events
with peak discharges ranging from 70.3 to 253 m³/s in the 2008-2012 period. The simulated discharge was compared to
USGS streamflow observations for all six gaging stations. We assessed the peak discharge and peak time to examine the


performance of the model. The GSSHA modeled and USGS gage measured hydrographs for three storm events are
compared in Fig. A2-A4 in Appendix A.
Peak discharge difference is calculated as the difference between the modeled peak and measured peak as a percentage of
the measured peak. Median peak discharge differences across all 21 events range from -35% to 57% (Fig. 2a). The largest
difference is in sub-basin DR-1. The reason is likely that DR-1 has a large area of land which was not represented fully on
county storm sewer maps (Smith *et al.*, 2015). The median peak discharge difference at the watershed outlet was -14%.
The peak time difference is calculated as the time difference between the simulated peak time and measured peak time (Fig.
2b). The median difference ranges from -15 to +10 minutes, which is within the temporal resolution of the data (15 minutes
for rainfall; 5 minutes for streamflow). The results show that the main tendency of flood response is captured by the model.
Overall, the validation shows that the physically-based, minimally-calibrated model can capture the main shape and timing
of the measured response in Dead Run. We therefore conclude that the model is suitable for the subsequent flood frequency
analysis. It should be noted that the errors in simulated response may be attributable to measurement errors tied to stage
discharge curves and to conversions of radar reflectivity to rainfall rate, as well as to the features that were simplified within
the model, such as initial soil moisture and some aspects of the storm drain network (Smith *et al.*, 2015).
**3.2 Flood frequency distribution**
Under the SST framework, 3-h rainfall scenarios for 10-yr, 50-yr, 100-yr and 200-yr return periods were generated (Fig.
A5 in Appendix A). We then simulated hydrographs using the GSSHA model and rainfall scenarios for Franklintown and
the five DR subwatershedss.
**3.2.1 Flow discharge estimates**
The distribution of maximum discharge at the Franklintown gage for rainfall return periods ranging
from 10 to 200 years is illustrated in Fig. 3a. To compare the distributions of rainfall and flood peaks,
the values are normalized to range from 0 to 1. The most striking feature is that the distributions of
total rainfall and flood peaks are highly variable across the four return periods. The kernel density
distribution of rainfall shows a peak at the position of $50^{th}$ quantile for four return periods. The
distribution of flood peak is more complex. For the 100-yr rainfall return period, the kernel density
distribution of flood peaks shows a multimodal trend with two small peaks around the $25^{th}$ and $75^{th}$
quantiles, which contrasts with the unimodal distribution of rainfall. For the 200-yr rainfall return
period, the interquartile range (IQR) is larger than other return periods. The relative standard deviation,



known as the coefficient of variation (CV), is used to present the dispersion of peak distribution. CV
is defined as the ratio of the standard deviation to the mean. Unlike the IQR results, CV decreases with
increasing return period. According to Zhou *et al.* (2019), the variability of basin-average total rainfall
increases with return period. The pronounced difference in the distributions of total rainfall and flood
peaks highlights a complex relationship between rainfall properties and flood response in this relatively
small urbanized watershed.
The flood response time is calculated as the difference between the time of maximum rainfall rate and
maximum discharge (Fi. 3b). Median values of response time are similar under all return periods,
ranging from 70 to 83 minutes, which, given the temporal resolution of rainfall is 15 minutes, can be
similar for all four return periods. It can be concluded that although the flood peak magnitude increases
with rainfall return period, the response time is consistent for various rainfall scenarios. This implies
in this small highly urbanized watershed the response time is more linked to the drainage system rather
than to rainfall characteristics.
Figure 4 demonstrates the simulated hydrographs for the four return periods. The upper and lower
spread (75th and 25th quantiles) of the hydrograph indicates the range of variability of simulated
hydrographs. For the 10-yr return period, the hydrograph is relatively smooth with smaller spread.
With increasing return period, the hydrograph is peakier with shorter duration of high magnitude
discharge. The hydrograph for the 50-yr return period shows a transitional shape between small (10-
yr) and large (100-yr and 200-yr) rainfall return periods. For the 100-yr return period, the upper spread
shows a tendency toward dual peaks, which cannot be revealed from conventional design flood
practices. For the 200-yr return period, the hydrograph is peakiest with a large upper spread.
**3.2.2 Spatial distribution of flood magnitude**
The distribution of flood peaks over the five subwatersheds exhibits contrasting variation with rainfall return periods
ranging from 10 to 200 years (Fig. 5). Generally, basin scale plays an important role in determining the distribution of flood
magnitudes. Under the 10-yr rainfall return period, DR1 and DR2, with basin scales of 1.3 and 2.0 km$^2$, have higher flood
peaks and interquartile ranges than other subwatersheds. DR5 (2.1 km$^2$) has comparable flood magnitude with DR4 (6.3
km$^2$) and Franklintown (14.3 km$^2$), while has a larger interquartile range than the latter two. DR3 with a basin scale of 4.95
km$^2$, has comparable flood magnitudes with DR1 and DR2. Under the 200-yr rainfall return period, DR2 and DR3 has a
slightly larger flood magnitude than DR1. DR5 has the largest interquartile range than others, though its flood peaks are





smaller than other small watersheds.
Results show that sub-basin flood distributions vary significantly with rainfall return periods. DR1 with larger impervious
area and dentention controlled area than DR2 (Table 1), has larger flood peaks under small rainfall return period. For large
return periods, DR2 has larger peak and interquartile range than DR1, implying that flood peaks are less impacted by
impervious area for extreme storms.   DR5, with the smallest dentention controlled area by detention infrastructure, has the
smallest flood peaks under small rainfall return period. Under large return period, however, it has the largest changes in
peak discharges with comparable flood peaks with subwatersheds larger than 6 km$^2$. DR3 and DR4, with basin scale of
4.95 and 6.29 km$^2$, have contrasting flood magnitude under small and large return periods. DR3 with larger impervious
area and dentention controlled area has larger flood peaks than DR4. The difference is more significant for small rainfall
events with the median value of flood peak for DR3 more than double that of DR4. From these results, it can be concluded
that impervious area and dentention controlled area play a significant role in determining the peak discharges, but the
impact reduces with increasing rainfall return period. The less dentention controlled sub-basin has larger flood variability
under large return period. The detention infrastructure impacts flood peak and its variability.
We further examine the spatial distribution of flood magnitude over the Dead Run watershed under the 100-yr return period
of flood at Franklintown (Fig. 6). The dimensionless flood index is used to compare flood peak magnitudes over the
watershed (Lu *et al.*, 2017). The flood index is computed as the maximum flow discharge divided by the computed 10-yr
flood ($Q_{10\text{-}y}$) at the same location, which is set as the median value of 10-yr peak discharge at the watershed outlet for each
100-yr design storm simulation. At Franklintown, the flood index and its interquartile range are largest across the
watersheds, with the median value greater than 2.5. The flood index in the five sub-watersheds is relatively lower, within
a median value between 1.5 and 2. DR2, as a sub-watershed of DR3, has a larger median value than DR1 and DR3. The
flood indices at DR1 and DR3 have similar median values and interquartile ranges. Values in DR4 are higher than its sub-
watershed, DR5, with a median value of 2. The variability of flood magnitudes, indicated by the CV, is stable among the
watersheds, ranging from 0.30 to 0.39. The spatial distribution of flood magnitude points to the significant heterogeneity
of flood distributions over the 14.3-km$^2$ watershed. For storm events that produce the same peak discharge return period at
the watershed outlet, the subsequent upstream flood response can vary substantially in the Dead Run watershed.
**3.3 Rainfall-Flood Relationships**
**3.3.1 Rainfall structure and flood response**
We investigate the relationship between the spatial and temporal characteristics of rainfall and flood response for small and
large rainfall return periods based on Spearman's rank correlation (Fig. A6 in Appendix A). The peak rainfall rate ($M_{max}$),





total rainfall ($R_{sum}$), fractional coverage ($Z$), rainfall location ($L$), rainfall-weighted flow distance ($RWD$) and the dispersion
of RWD ($S$) are used to characterize rainfall spacetime structure. For the 10-yr return period, the flood peak is somewhat
correlated with total rainfall, peak rainfall rate and storm core coverage with correlation coefficient of 0.16. For the 200-yr
return period, in contrast, there is no significant correlation between these features with correlation coefficients of -0.09,
0.07 and -0.02, respectively, implying a complex and nonlinear relationship between extreme storms and floods in the
watershed.
We used random forest regression models to examine the importance of rainfall characteristics to the flood response.
Rainfall spacetime structure characteristics are used as RF model features. The flood peak is set as the model target. The
main parameters of RF model are tuned by a grid search approach(Probst *et al.*, 2019). The prediction performance is
assessed using Mean Absolute Error (*MAE*), Root Mean Square Error (*RMSE*), and explained variance regression score (*E*
score)(Achen, 2017). Smaller values of *MAE* and *RMSE* indicate better model performance. *E* score ranges from 0 to 1 and
a larger value indicates a better model (The training process of RF model is shown in Fig. A7 in Appendix A). The difference
in feature importance is compared between the 10-yr and 200-yr return periods (Fig. 7). For the 10-yr return period, peak
rainfall rate ($M_{max}$) and total rainfall ($R_{sum}$) are the most two important features. For the 200-yr return period, however, the
dispersion of RWD ($S$) and fractional coverage of storm core ($Z$) are more important than peak rainfall rate and total rainfall.
The rainfall location ($L$) has the smallest importance for both return periods. The results demonstrate the different
relationships between rainfall structure and flood response under small and extreme rainfall events. For extreme storms,
the maximum discharge is more closely linked to the spatial structure of rainfall, which is consistent with the results in
(Peleg *et al.*, 2017; Zhu *et al.*, 2018).
The temporal shapes of hydrographs and hyetographs are compared by using the coefficient of skewness (Fig. 8). The
skewness is used to assess the shape of rainfall process and discharge process. A negative value of skew indicates a left tail
of the distribution, and positive indicates a right tail. For the 10-yr return period, the rainfall skewness ranges from -0.1 to
3.5, demonstrating the mixed shapes of temporal distribution. Similar features are found for discharge shapes. For the 200-
yr return periods, the skewness of discharge is mostly positive while the skewness of rainfall events still varies from -1 to
2.5. The general conclusion of these analyses is that regardless of the temporal distribution of rainfall, the flood response
is relatively rapid, highlighting the role of the urban drainage system for the hydrographic response. The relationship
between the variability of discharge and rainfall is not significant for the four return periods, which implies that in a highly-
urbanized watershed, the drainage system smooths rainfall variability somewhat.





### 3.3.2 Rainfall return period vs. flood return period

In conventional design storm/flood practices, the return period of rainfall and peak discharge is often assume to be equivalent (Rahman *et al.*, 2002). Under the SST framework, we can examine this assumption (Wright *et al.*, 2014a). For each SST realization containing 100 rainfall scenarios with return period from 5 years up to 100 years, the peak discharge can be simulated and ordered. Flood frequency for return periods from 5 years up to 100 years are then estimated from the ordered peaks. We run 30 SST realizations in total. The Spearman's rank correlation of the two return periods is 0.5 (Fig. 9). The results quantitatively confirm that the assumption of a 1:1 return period equivalency between design storm and design flood cannot hold, even in a small highly-urbanized watershed where drainage network and rainfall structure play an important role in flood response.

### 3.3.3 Impact of rainfall spatial heterogeneity on flood responses

We also compared the simulated flood response resulting when rainfall is uniform over the watershed, rather than spatially distributed as in previous analyses (Fig. 4 and Table 2). Generally, the flood peaks generated from uniform rainfall have lower peaks than for non-uniform rainfall. The difference increases with return period. Under the 10-yr return period, the shapes of the two hydrographs have similar upper and lower bounds (75% and 25% quantiles). The median flood peak using non-uniform scenarios is 22% higher than the uniform scenarios. Under the 200-yr return period, the hydrograph resulting from non-uniform rainfall is much peakier than the uniform SST scenarios with higher upper and lower bounds. The lower bound of hydrograph by non-uniform SST scenarios is close to the median hydrograph of uniform SST scenarios. The impact of rainfall spatial heterogeneity among the five subwatersheds is different. DR1, with a basin scale of 1.32 km$^2$ and located in the north-west boundary of the watershed, was the least-impacted by rainfall spatial distribution for all return periods. In DR2, on the other hand, which is similar in drainage area to DR1, the flood peak increased by 46% for the 200-yr return period. For DR3 and DR4, the spatial heterogeneity of rainfall contributes more to the flood peaks in DR4 than in DR3. The most striking difference in flood peaks is in DR5 for the 50-yr return period. The difference in flood magnitude is 75%. As mentioned above, DR5 is the sub-basin with the least dentention controlled area. This finding is likely tied to the complex relationship between space-time rainfall structure and the drainage network. We can thus conclude that the spatial heterogeneity of rainfall can increase flood peaks dramatically under both small and large return periods. The impact increases with return period. This result shows that the assumption of spatially uniform rainfall will underestimate flood frequency.

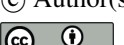

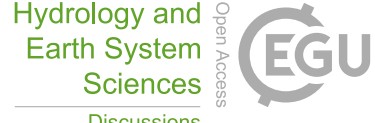

**4. Summary and conclusions**

This paper addresses the problem of the impacts of short-duration rainfall variability on hydrologic response in small urbanized watershed. By coupling a high-resolution radar rainfall dataset and stochastic storm transposition (SST) with the GSSHA distributed physics-based model (see also (Wright *et al.*, 2014a; Zhu *et al.*, 2018), the relationships between rainfall spatiotemporal structure and urban flood response is examined. The main findings are as follows:

1. The flood frequency distributions for subwatersheds within the highly-urbanized 14.3 km$^2$ Dead Run watershed demonstrates the complexities of flood response for both short and long rainfall return periods. Especially for 3-h extreme storms, the distribution of flood peaks shows large variability. The variability of flood magnitude shows a pronounced role of rainfall space-time structure in flood production. This calls into question the commonly-made design storm assumption of spatially uniform rainfall. The response time is less affected by rainfall structure and appears to be more closely associated with the basin scale and drainage network features.

2. The spatial heterogeneity of flood frequency over the 14.3-km$^2$ watershed is striking for the 100-yr return period. The intercomparison between subwatersheds show that the impact of impervious area decreases with increasing return periods. The subbasin with the least detention infrastructure shows the largest flood variability for long return periods. For the100-yr return period, the flood index of five subwatersheds are different from that of their downstream outlet. It shows that storm events that produce the same peak discharge return period at the basin outlet can be the result of very different upstream flood responses.

3. The relationship between the spacetime structure of rainfall and flood response is complex. The random forest-based feature importance analysis shows very different relationships between rainfall structure and flood response for frequent vs. extreme rainfall events. For smaller and more frequent rainfall events, flood peaks are more closely linked to the temporal features of rainfall (total rainfall and peak rainfall rate). For extreme storms, the maximum discharge is closely linked to the spatial structure of rainfall (storm core coverage). This finding is broadly consistent with (Peleg *et al.*, 2017) and (Zhu *et al.*, 2018), despite the very different drainage scales considered in those studies. There is no significant correlation between rainfall peak, total rainfall and flood peaks, implying an important role of surface properties in urbanized watersheds. Similar to (Wright *et al.*, 2014a), this comparison calls into question the conventional design storm assumption of a 1:1 equivalency between rainfall and flood peak return periods.

4. The spatial heterogeneity of rainfall is a key driver of flood response across scales. Relative to spatially uniform rainfall, spatially distributed rainfall can increase flood peaks by 50% on average at the watershed outlet and its subwatersheds for both small and large return periods. This finding is broadly consistent with prior results at much larger scales in an



agricultural setting ((Zhu *et al.*, 2018)) and suggests both spatial and temporal rainfall distributions need to be considered
in flood frequency analyses, even in relatively small urban watersheds. This study also implies that the drainage network
substantially alters the impact of rainfall characteristics on the runoff.
Coupling the GSSHA model and SST-based rainfall frequency analysis, this study provides an effective approach for
regional flood frequency analysis for urban watersheds. It can be used to explore the dominant control on the upper tail of
urban flood peaks, without many of the limiting assumptions associated with design storm methods. The study area could
be extended in future work with larger basin scales and by manipulating the spatial heterogeneity of basin characteristics
within GSSHA or other similar modeling systems.
**Acknowledgments**
This study was supported by the National Science Foundation of China (Grant 51909191).
**Data availability**
Radar data are archived at Princeton University and can be downloaded from the url
http://arks.princeton.edu/ark:/88435/dsp01q524jr55d.

**Author contributions.**
Main contributions from each co-authors are as follows. Zhengzheng Zhou contributed to computation and organization of
the paper. James A. Smith contributed to the supervision and writing. Mary Lynn Beack is responsible for generating the
radar rainfall data. Brianne K. Smith contributed to the construction of the initial hydrological model. Daniel B. Wright
contributed to the writing of the paper. Shuguang Liu contributed to the supervision and writing.

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

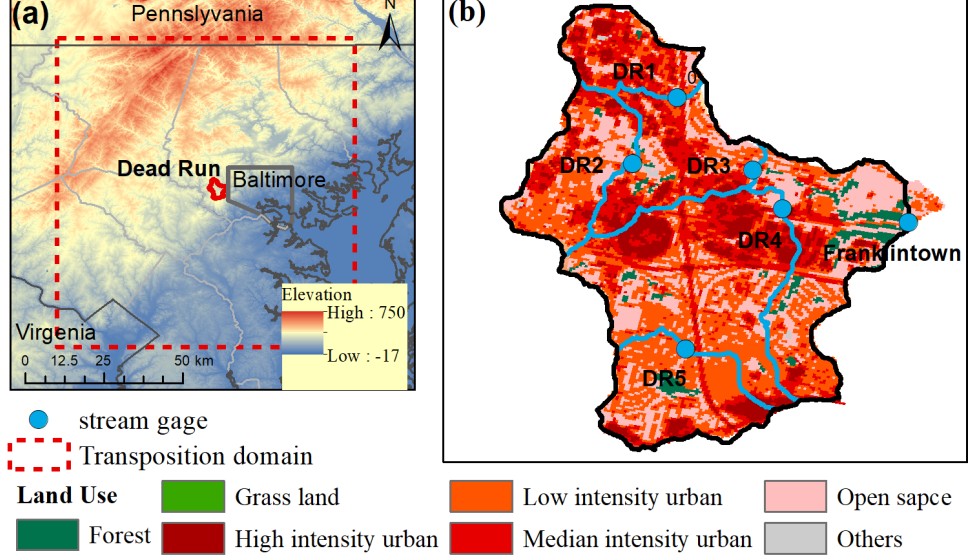






**Figure 1. Overview of Dead Run study region including (a) location of DR, elevation, and transposition domain of**
**SST; (b) land use land cover and stream gages. Land use land cover was obtained from the National Land Cover**
**Data set (NLCD, http://www.mrlc. gov)**

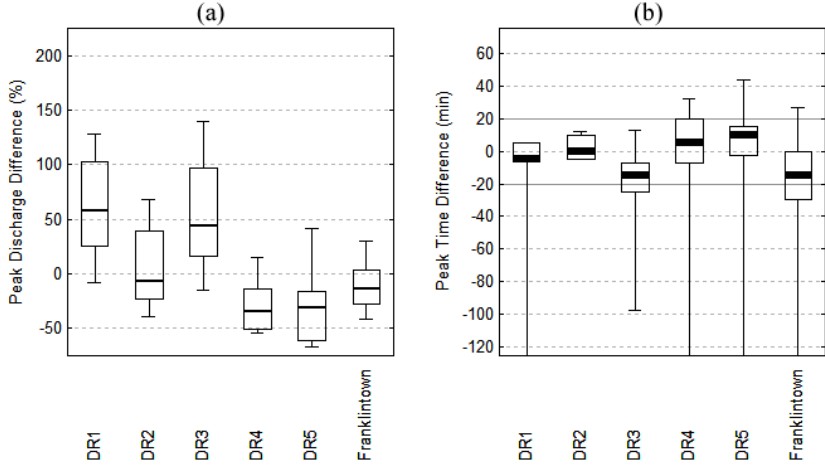


**Figure 2. Comparison of (a) flood peak discharges and (b) response times for 21 historical rainfall events.**

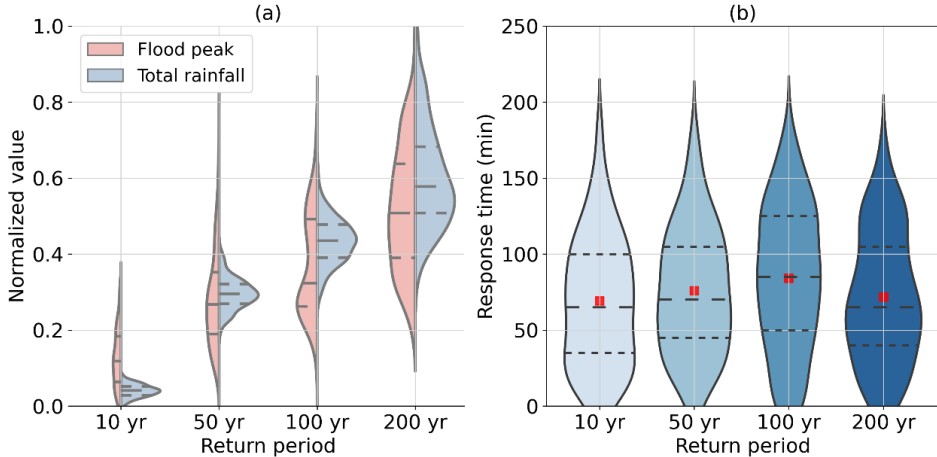


**Figure 3. Violin plots of (a) normalized flood peak and normalized total rainfall; and (b) response time based on the**
**3-h design storms from 10-y to 200-y return periods. (The red dot indicates mean value. Dashed line in the middle**
**indicates the median value. Upper and lower dashed lines indicate the 75th and 25th quantiles, respectively.)**



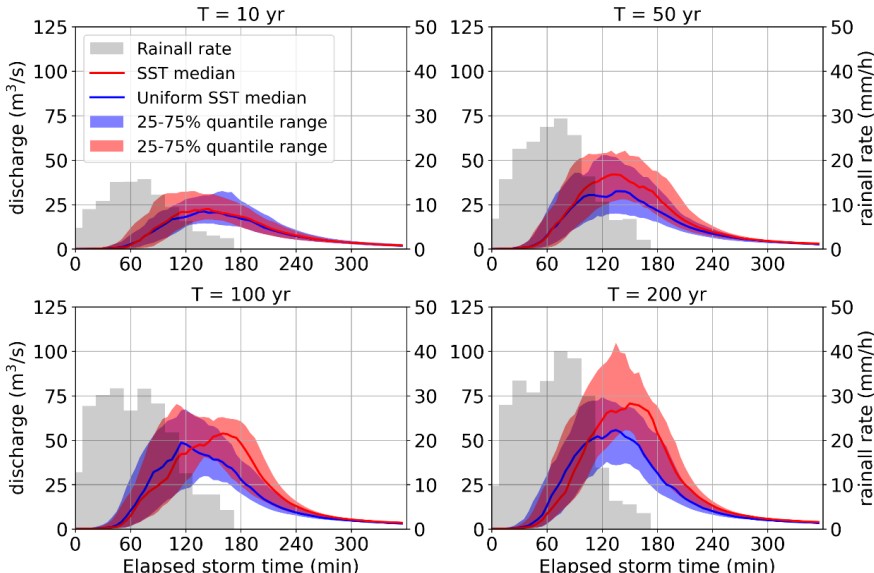


**Figure 4. Time series of simulated hydrographs for Franklintown based on the 3-h design storms from 10-yr to 200-yr return periods with spatially uniform (blue) and spatially distributed (red) rainfall. The grey bar indicates the median value of basin-averaged rainfall rate.**

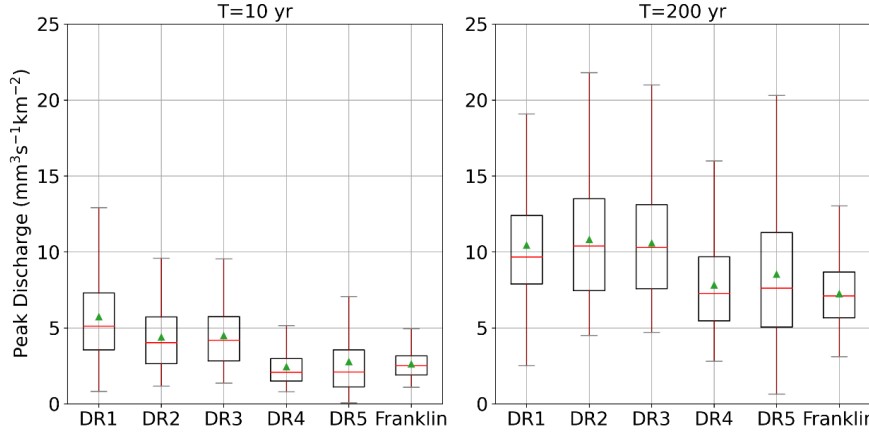


**Figure 5. Boxplots of normalized flood peaks for Franklintown and five subwatersheds.**



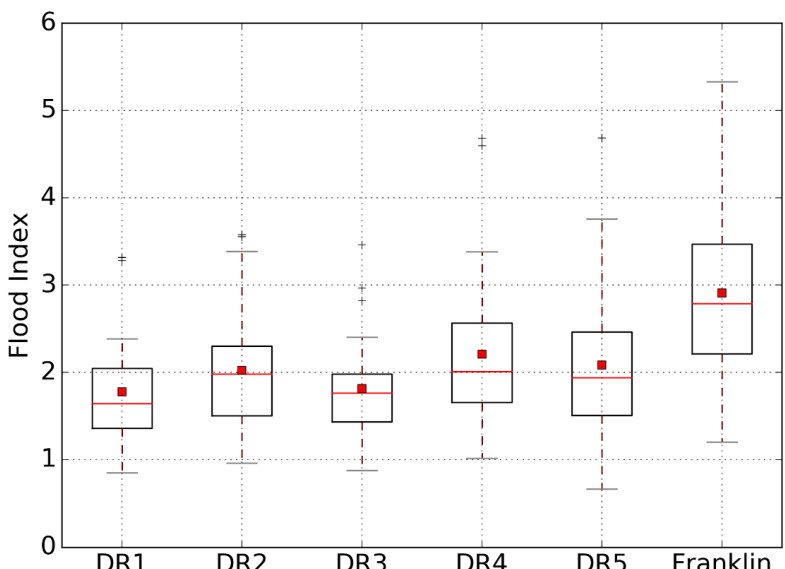

**Figure 6. Boxplot of flood index across the DR subwatersheds for the 100-yr design storms.**

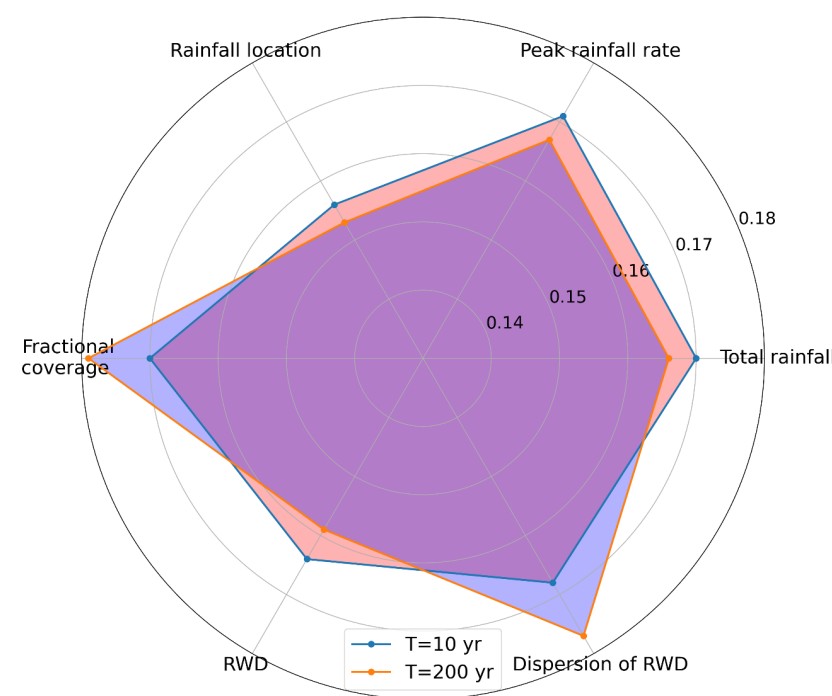

**Figure 7. Feature importance analysis of RF model for space-time rainfall structure and 10-yr (red) and 200-yr (blue)**
**flood peaks.**





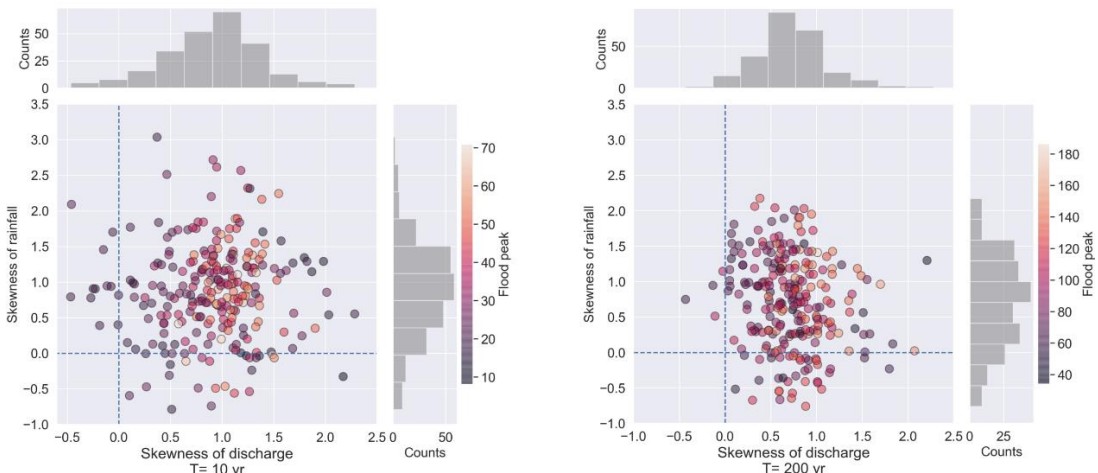


**Figure 8. Scatter plots of skewness of rainfall and peak discharge—left: 10-yr return period; right: 200-yr return**

**period.**

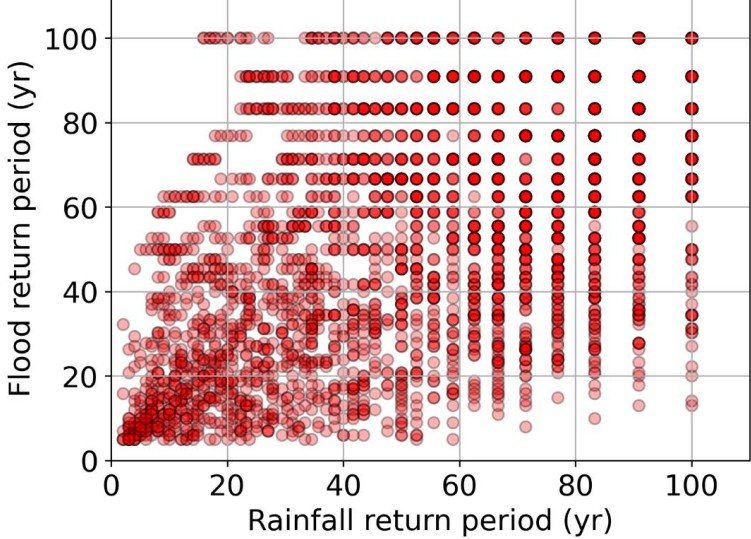


**Figure 9. Scatterplot comparison return periods for rainfall and peak discharge for individual SST-based**
**simulations.**

**Table 1: Characteristics of Dead Run watershed.**

|  | USGS ID | Area (km²) | Developed Land (%) | Imperviousness (%) | Controlled area (%) |
|---|---|---|---|---|---|
|  |  |  |  |  |  |





| | | | | | |
|---|---|---|---|---|---|
| DR1 | 01589317 | 1.32 | 99% | 73.6 | 41.9 |
| DR2 | 01589316 | 1.92 | 98% | 55.5 | 18.5 |
| DR3 | 01589320 | 4.95 | 98% | 62.2 | 24.4 |
| DR4 | 01589315 | 6.29 | 98% | 51.5 | 12.2 |
| DR5 | 01589312 | 2.05 | 96% | 47.9 | 3.2 |
| Franklintown | 01589330 | 14.3 | 96% | 52.3 | 25.1 |

**Table 2. The median flood peak reductions using spatially uniform and spatially distributed**
**rainfall.**

| | T=10 yr | T=50 yr | T=100 yr | T=200 yr |
|---|---|---|---|---|
| DR1 | 14% | 20% | 13% | 26% |
| DR2 | 19% | 40% | 28% | 42% |
| DR3 | 24% | 33% | 27% | 31% |
| DR4 | 32% | 51% | 38% | 35% |
| DR5 | 15% | 75% | 37% | 30% |
| Franklin | 22% | 36% | 31% | 42% |
