# Peer review of "The impact of spatiotemporal structure of rainfall on flood"

_Hydrology and Earth System Sciences, 2021_

## Author Comment (AC1)

**Response to Anonymous Referee #1**

Comments/text of reviewer posted in **bold** *and italics*; the authors's answeres start with "*Response*:"; the sentences in the revised version is in blue.

**Overall remarks**

*In their paper the Authors have examined the role of rainfall spatial and temporal variability in flood frequency across drainage scales in the highly-urbanized Dead Run watershed (14.3 km2) outside of Baltimore, Maryland, USA with the use of a flood frequency analysis framework that combines stochastic storm transposition-based rainfall scenarios with the physically-based distributed GSSHA model. The results they obtained show the complexities of flood response within several subwatersheds for both short (< 50 years) and long (> 100 years) rainfall return periods. The Authors revealed that the impact of impervious area on flood response decreases with increasing rainfall return period and for extreme storms, the maximum discharge is closely linked to the spatial structure of rainfall, especially storm core spatial coverage. The spatial heterogeneity of rainfall increases flood peak magnitudes by 50 % on average at the watershed outlet and its subwatersheds for both small and large return periods. According to the Authors, the results imply that commonly-made assumption of spatially uniform rainfall in urban flood frequency modeling is problematic even for relatively small basin scales.*

*I deeply admire the effort the Authors made while preparing their article. I have found it very interesting and thought provoking. In my opinion the paper is relatively well written, presents interesting and appealing (from the practical point of view) approach to the analysis of rainfall-runoff processes in small urban catchments, and it may be inspirational for scientists performing similar analysis in other cities. I am also pleased to say, that the paper was written with care. Here, it this review let me just concentrate on some issues that I believe, could be corrected.*

*Heaving read the paper, my first impression was that, however interesting, the paper is a bit too wordy in some parts (e.g. introduction or discussion) and its content would be 'squeezed' by a page or two.*

*The first chapter roughly describes the problem of the translation of rainfall spatiotemporal distribution into flood responses. In their work the Authors concentrate on small urban area advocating that due to the complexity of the hydrologic and rainfall spatialtemporal conditions on flood frequency analysis in smaller urban areas still provides a room for scientific investigation. Although the Authors noticed that this is not a new scientific issue, they claim that the novelty of their research stem, inter alia, from better understanding of spatial and temporal diversity of rainfalls and generation of runoff which is now possible because of modern techniques of rainfall monitoring and modelling of hydrological processes. They pointed out also that the problem of flash inundations in especially small urban areas has not received proper attention among the researchers. The Authors present exhausted literature review in this field, however some of the papers they cited are not new and perhaps the whole list could be completed by newer research results.*

*Response:* We would like to thank the reviewer for the detailed and thoughtful review. We have revised throughout the paper based on your comments. We have re-organized and shortened some paragraphs in the introduction and results. About 120 words in the introduction and 700 words in the results are deleted. In the introduction in the revised version, some early studies (such as Saghafian et al., 1995; Dawdy and 32 Bergmann, 1969; Schilling, 1991) are replaced by newer findings (such as Yin *et al.* (2016); Yang *et al.* (2020); ten Veldhuis *et al.* (2018)).

*In order to improve the modelling of the spatio-temporal rainfall conditions in the small cachments the Authors apply the Stochastic Storm Transposition method which combined with hydrological models can be used for multiscale rainfall frequency analysis and flood frequency analysis.*

*In this chapter, the Authors state two scientific questions (page 3, lines 69-72): '(1) How does flood frequency in small urban watersheds vary with diverse space-time rainfall structure and rainfall magnitude? (2) Among the space-time feature of rainfall, what are the dominant features that control flood peak distribution in small urban watersheds?'*

*In my opinion these questions cannot contribute to the development of the hydrological sciences but deal rather with the use of already fossilised knowledge on the rainfall-runoff processes, especially in relatively poorly recognised conditions of a (just one) small highly urbanised area. I suggest to rephrase the main goal of the research in order to emphasise a new approach to solving the problem of modeling the processes of flooding in urbanized catchments within more general context.*

*Responses*: Thank you for your suggestion. The SST-based flood frequency analysis with radar rainfall data was proposed and applied by Wright *et al.* (2014a) in 2014. So to develop a new approach for flood frequency analysis is not the main novelty of this study.

"In the conventional frequency analysis, several idealized assumtions include idealized rainfall temporal structure with unimodal peak, uniformed spatial distribution and 1:1 rainfall-flood return periods. These assumtions ignore the interaction between spatiotemporal structure of rainfall and flood responses, which increases the uncertainty of frequency estimations." Thus, we argrue that "the influence of rainfall spatial-temporal structure on flood frequency analysis in urban areas remains an open research issue" in the first paragraph in Section1.

On top of that, the main contribution of this paper is to explore the rainfall variability and its impact on quick flood response in a small-scale urban watershed. The two questions are raised to explore the rainfall-flood interactions in a small urbanized watershed. We have found the different impacts of rainfall spatiotemporal features on flood responses under small and large return periods in this study. These results provide insights for small-scale flood frequency analysis.

In the conclusion, we still addressed the value of the new-developed flood frequency analysis as follows: "Coupling the GSSHA model and SST-based rainfall frequency analysis, this study provides an effective approach for regional flood frequency analysis for urban watersheds."

*The second chapter presents the data and the case study area which is the highly urbanized 14.3 km² Dead Run (DR) watershed located west of Baltimore, Maryland, USA. The Authors chose this area because its wealth of data is exceptional to examine rainfall and hydrologic response, and some preliminary research was already performed in this region, too. It is to me unclear why the Authors decided to carry out their research on watershed rather than within a catchment which is easier to model because of water balance relationships. As an input to the models the radar-detected rainfall data corrected with the use of 54 rain gauges in and around Baltimore City were applied. However, the authors refer the reader to the publication by Zhou et al. (2019) for the detailed description of the methodology of bias correction, it would be good if the Authors could shortly describe these techniques in their paper within a paragraph or two which might improve coherence of the publication as well as underline its highlights.*

*Responses*: Dead Run is a tributary of the Gwynns Falls watershed which is the principal study watershed of the Baltimore Ecosystem Study (BES; see Groffman *et al.* (2003); Smith *et al.* (2005) ) and has a

dense network of stream gauges that have been deployed in connection with BES hydrologic monitoring activities. With more data, the high-resolution radar rainfall data can be corrected more precisely, and the small-scale flood response can be examined. Because of its wealth of data, numbers of hydrological research have already been performed in the watershed, showing that the series of relevant studies demonstrate the modeling in DR is effective and reasonable for hydrological studies.

We have rephrased the paragraph and added a general introduction of bias correction for radar data. In the revised version, it is written as follows. "The Hydro-NEXRAD algorithms includes quality control algorithms, *Z-R* conversion of reflectivity to rainfall rate, time integration, and spatial mapping algorithms (Seo *et al.*, 2011). To improve the rainfall estimates, a multiplicative mean-field bias correction (Smith and Krajewski, 1991; Wright *et al.*, 2012) is applied on a daily basis using a network of 54 rain gauges in and around the Baltimore County. The bias computation takes the form $B_i = \frac{\sum_{S_i} G_{ij}}{\sum_{S_i} R_{ij}}$.

Where $G_{ij}$ is the rainfall accumulation for gage *j* on day *i*, $R_{ij}$ is the daily rainfall accumulation for the co-located radar pixel accumulation on day *i*, and $S_i$ is the index of the rain gage stations for which both the rain gage and the radar report positive rainfall accumulations for day *i*. Each 15-min radar rainfall field from day *i* is then multiplied by $B_i$."

***The discharge data for six gages located in the case study area with a resolution of five minutes from the U.S. Geological Survey (USGS) were used for identification of the runoff model parameters. The problem is, however, that only one of the six gauges provides reasonably long datasets of discharges, the rest 5 provide data from 2008, which in my opinion is too short to observe any long-term alterations in rainfall-runoff regime of this region, especially in the area where the intensive urbanisation processes have been occurring. Moreover, the dataset for such a short period should not be applied in the FFA.***

*Response:* The long-term alteration in rainfall-runoff regime in a developing area is beyond the scope of this study. The focus of this paper is to investigate the impact of the spatiotemporal structure of rainfall on the flood responses in a highly urbanized watershed (DR with a developed area of 96%) under different rainfall return periods. Nonetheless, we stress that a number of previous studies have demonstrated that the combination of relatively short (i.e. 10-20-year records) of bias-corrected radar rainfall with SST can, by virtue of SST's space-time substitution, yield fairly reliable estimates of rainfall and flood frequencies out to 100-1,000 years (see Wright *et al.* (2013); Wright *et al.* (2014a); Wright *et al.* (2017); Yu *et al.* (2019)). The first three of these papers discuss the issue of space-time substitution via SST extensively. In this sdtuy, the high-resolution radar rainfall data is from 2000 to 2015 with a record of 16 years, which can provide reasonable rainfall frequency results.

Furthermore, the discharge at Franklintown shows stationarity from 2015 to present (Figure R1), demonstrating that the model can present the current discharge condition in the watershed.

The framework used in this study provides an alternative approach for FFA. The approach can be applied to other regions with longer rainfall and discharge records. We believe that with increasing data records, the results can be improved continuously.

[Figure]

**Figure R1**. The daily discharge at Franklintown for the 2000-2021 period.

*Section 2.2 describes the two-dimensional GSSHA model to simulate multi-scale flood response, whose structure for the Dead Run was created by one of the Authors in 2015 and later modified for the purpose of the new research. The rainfall scenarios were shortly described in section 3.3. The RainyDay, an open source SST software package was used for this purpose. From the text one can infer that the Authors use ready-to-use models and techniques to perform their case study research. Obviously, these techniques might have needed some modifications and adjustments to the specifics of the Dead Run region, but it is not clearly stated in the paper whether these modifications go beyond the regular adaptation of the models to the case study.*

*Response*: This study is a continuation of Zhou *et al.* (2019) in DR watershed, which developed an intensity factor for transposed storm magnitude. The rainfall results for DR watershed are directly obtained from Zhou *et al.* (2019). The GSSHA model is slightly modified based on the DR model in Smith *et al.* (2015). The modification validation of the model is discussed in Section 2.2 and Section 3.1, showing that "the physically-based, minimally-calibrated model can capture the main shape and timing of the measured response in Dead Run. We therefore conclude that the model is suitable for the subsequent flood frequency analysis."

*The 2.4 subsection depicts the characteristics of rainfall and hydrologic response. The methodology presented in this section suggest averaging the rainfall over the whole modelled area (Eqs. 1 -3) which in my opinion leads to the averaging the results and ruins the spatial diversity of the rainfall events and the local catchment responses characteristic for urban area with highly differentiated land cover. The methodology presented in this subsection deserve more comment.*

*Response*: We agree that the basin-averaged index will ignore the potential spatial heterogeneity over the watershed. First, we will point out that with the exception of Section 3.3.3, all model simulations were done using spatially-distributed rainfall. With the exception of that Section, we use basin-average rainfall only as a descriptor, along with others that such as peak rainfall rate, fractional coverage, etc. Taken together, these descriptors help us tease out which aspects of rainfall are most important in determining flood response.

We have added comments in the revised version as follows. "The Eqs.1-3 are typical rainfall characteristics used in conventional rainfall-flood analysis since they reflect the general information of rainfall. Since the basin-averaged index will ignore the potential spatial heterogeneity over the watershed, Eqs. 4-8 describe the spatial distribution of rainfall within the area. "Our results demonstrates that there is less correlation between the basin-averaged rainfall features and flood responses for large return periods.

*Chapter 3 discusses the results of the simulations. Unfortunately it is hard to assess the accuracy of the models calibration, because they are described in appendix A which I could not find.*

*Response***:** We have added Figure 3 in the revised version as an example of hydrograph for the 14 August 2011 storm event to show the performance of the hydrological model. The other hydrographs are demonstrated in the appendix at the end of the revised version.

[Figure]

**Figure 3. Hydrographs and rainfall for the the 14 August 2011 storm event. Time refers to minutes from the start of the model simulation.**

*However, the Authors admitted (graphs in Fig 2) that the differences of the estimation of the peak flows range from -35 to 57% of the peak hight (probably in m3/s) which means that the models used are technically useless, even though they perform reasonably good for average discharges.*

*Response***:** The flood magnitude at the downstream Franklintown gage is well captured with the median peak discharge difference of -14%. For the subbasins, DR-1, as an instance, has the largest median peak discharge difference of 57% because it has a large area of land which was not represented fully on county storm sewer maps (Smith *et al.*, 2015). A similar situation is found in Smith *et al.* (2015), and their research showed that the model can be effective in producing hydrological responses. Overall, for such a large collection of flood events with various rainfall characteristics and peak discharges ranging from 70 $m^3$/s to 253 $m^3$/s, the model performs well in reproducing peak discharges. We have revised the discussions of the accuracy of model simulation over the watersehds.

*The temporal modelling results are closer to reality and differ only by a quarter from the actual peak time. The misleading estimation of the flood magnitude influences the conclusions presented in the paper. Perhaps use of other modelling tools or better identification of parameters would improve the simulation results. Also the quality of data would affect the mistakes. Obviously, the radar-based rainfall data cannot achieve the accuracy compared to on-ground pluviometric monitoring service.*

*Response***:** In the study of Smith *et al.* (2015), the DR model performed well in simulating flood responses under various rainfall storms. We thus still use the model in our study with only minor modifications.

The median difference of peak-to-peak response time ranges from -15 min to +10 min, which is within the temporal resolution of the data (15 min for rainfall; 5 min for streamflow). The median Nash-Sutcliffe Efficiency (NSE) also shows the model performs well, especially at Franklintown (the median NSE is 0.77 (Fig. 2c)).

We agree with the reviewer that the error in conversions from radar reflectivity to rainfall rate may also impact the model output. The error in simulated response may be attributable to measurement errors tied to stage-discharge curves and to conversions of radar reflectivity to rainfall rate, as well as to the features that were simplified within the model, such as initial soil moisture and some aspects of the storm drain network (Smith et al., 2015). We feel that the magnitude and timing of flood response is broadly captured by the model, especially at the downstream gage.

The discussion of error from discharge is added in the revised version as follows. "It should be noted that the error in simulated response may be attributable to measurement errors tied to stage discharge curves and to conversions of radar reflectivity to rainfall rate, as well as to the features that were simplified within the model, such as initial soil moisture and some aspects of the storm drain network (Smith *et al.*, 2015). For example, it has been documented that the average error of discharge between USGS direct measurements and stage-discharge curves for Franklintown is 17.4% between 2008 and 2010 (Lindner and Miller, 2012). As mentioned in Section 2.1, the rating cruve used to compute the discharge data at DR3 and DR4 is not provided from USGS. It may increase the error in the measurements and modeling results. For the rainfall data set used in this study, the difference of the storm total rainfall between a rain gage and the bias-corrected radar rainfall data for the pixel of that gage is compared. The median difference for all gages over the 21 storms is 22.6%(Smith *et al.*, 2015)."

***The Authors claim that the rainfall return periods were calculated by mean of nonparametric kernel function method for periods up to 200 years. It is not clear what dataset (a few-years-long measurements?) were used for this estimation and what parameters of the kernel function were applied.***

***Response***: In Figure 4 in the revised version, the kernel density is used to demonstrate the distribution of rainfall totals results under different return periods. Under the SST framework, the ordered "annual" maxima with return period up to 200 years can be synthesized through storm transposition procedure. In this study, the recreation step is repeated 300 times and 300 such realizations of 200-yr series are generated. Thus, in Figure 4, the violin plot of 3-h total rainfall for the four return periods shows the kernel density distribution of the 300 realizations of the 200-yr series.

[Figure]

**Figure 4. Violin plots of (a) normalized flood peak and normalized total rainfall; and (b) response time based on the 3-h design storms from 10-y to 200-y return periods. (The red dot indicates mean value. Dashed line in the middle indicates the median value. Upper and lower dashed lines indicate**

*The comments based on this model's results are either trivial e.g. 'For the 200-yr rainfall return period, the interquartile range (IQR) is larger than other return periods.' or at least strange 'Unlike the IQR results, CV decreases with increasing return period', as the uncertainty and thus variability grows rather with the return period of the estimated quantile, in any catchment.*
**Response**: We have removed the argument about CV. The purpose of the comparison of rainfall and flood is to highlight a complex relationship between rainfall properties and flood response in the small urbanized watershed.

*In my opinion, however interesting, the results obtained for the Dead Run case study cannot be easily generalised for the similar catchments even though one used the same models and techniques. I would suggest to compare the results with other similar catchments to make it more universal. Otherwise the paper would attract only local interest.*
**Response**: The study aims to examine the role of rainfall space-time structure in flood response in small highly-urban watersheds. The methodology of SST combining hydrological numerical model addresses the new approach for a more detailed rainfall/flood frequency analysis. The results demonstrate the complex relationship between rainfall spatiotemporal structure and flood responses and highlight that the assumption of a 1:1 return period equivalency between design storm and design flood cannot hold even in a small watershed. The results challenge the conventional approach and provide an important reference for design storms in urbanizing areas. In the future, we will analyze more regions with different basin scales and climate features (that is what we are doing at present).

*Having read the discussion I could not resist the impression, that the Authors would like to analyse too many complex phenomena by means of (too) short and insufficient data and tools with serious limitations of their use. As a result the simulations are stricken by large uncertainty or evident mistakes (as with the CV-quantile return period relation).*
**Response**: We have explained the issue of data length, model error and variation of flood simulation (CV) in the above resonses. About the data length, with increasing records of radar rainfall data, the results can be improved. As mentioned before, however, SST has been shown in previous studies to be an effective method for "lengthening" records via space-time substitution of high-resolution extreme rainfall observations.

*Perhaps concentrating on one event in one place (e.g. catchment response to the torrential rainfalls) would account for the quality of the paper.*
**Response**: Thank you for the suggestions. Extreme rainfall case studies have been examined in (Smith *et al.*, 2013; Smith *et al.*, 2016; Zhou *et al.*, 2019) and many other places. Generalization beyond individual events to return periods has been much less studied. Here, the model was run for a large number of flood events relative to previous modeling studies to ensure that we captured a wider diversity of flood responses to differing rainfall events.

*On the other hand, when almost all input parameters are in fact modelled (e.g. rainfall data based on radar measurements, 200-year quantiles based on short datasets) one cannot expect credible accuracy*

*of the results. On top of that, poorly estimated models generate some extra bias additionally increasing the uncertainty of the results.*

**Response**: In conventional rainfall frequency analysis, there are also several simplified assumptions including idealized area reduction factors (ARFs) and temporally idealized design storms. In flood frequency analysis, the assumption of 1:1 return period equivalence between rainfall and modeled discharge is widely used. Therefore, these assumptions and simplified methods in conventional approaches can also increase the uncertainty of results.

Broadly speaking, with such a high-resolution rainfall dataset, physically-based model and SST framework, the approach yields advantages relative to other flood frequency analysis approaches, especially in urban areas. These advantages are discussed in (Wright *et al.*, 2014b; Zhuang *et al.*, 2020).

*The last chapter concludes the paper. This chapter summarises the text in a concise way in the form of pin-points but I lack any reference to the universality of the obtained results. I would expect any 'take home' recommendation for other hydrologists and practitioners. I am not also sure whether Authors managed to provide responses to the two questions stated in the first chapter.*

**Response**: The two questions in the Introduction are: (1) How does flood frequency in small urban watersheds vary with diverse space-time rainfall structure and rainfall magnitude? (2) Among the space-time feature of rainfall, what are the dominant features that control flood peak distribution in small urban watersheds? The responses to the questions are discussed in Section 3 Results. We summarized the complexities of rainfall-flood relationship under varies rainfall return periods which is the answer to the first question. To the second question, we summarized that for frequent rainfall events, flood peaks are more linked to the temporal features of rainfall, while for extreme storms, the maximum discharge is linked to the spatial structure of rainfall. The basin scale and urbanized drainage network features also increase the complexity of the rainfall-flood responses.

Thus, the results highlight the uncertainty of the conventional design storm assumption of a 1:1 equivalency between rainfall and flood peak return periods. It is also suggested that both spatial and temporal distribution of rainfall need to be considered in rainfall-flood frequency analysis. These are very important references for not only hydrologists but also engineers and planners.

Coupling the GSSHA model and SST-based rainfall frequency analysis, this study provides an effective approach for regional flood frequency analysis for urban watersheds. The study area could be extended in future work with larger basin scales and by manipulating the spatial heterogeneity of basin characteristics within GSSHA or other similar modeling systems.

**Specific comments**

*The Authors refer to the Appendix A, which I could not find.*

**Response**: The supplementary file is uploaded as a separate file in the submission system. In the revised version the appendices are attached at the end of the revised version.

*Map in Fig 1. is unreadable. What are the blue lines? Where is the Dead Run watershed? Misspellings in the map's legend.*

**Response**: The blue lines are the outlines of sub-basins boundary. The DR watershed with the red boundary is located in the center of the transposition. The captain of the figure is modified. The misspelling is corrected in the revised version. Thank you.

[Figure]

**Figure 1. Overview of Dead Run study region including (a) location of DR, elevation, and transposition domain of SST; (b) land use land cover and stream gages. The red outline and grey outline in (a) indicates the boundary of DR watershed and Baltimore City, respectively.**

*Figure 3 is incoherent.*

*Response***:** We have explained the kernel density functions and the distribution of rainfall-flood results in the above questions. Thanks.

*Technical remarks*

*The size of fonts varies throughout the manuscript.*

*Response***:** We have revised the font sizes throughout the manuscript. Thanks.

*Very difficult to assess the results when graphs and tables are attached at the end of the file instead in their original place (scrolling necessary, or printing).*

*Response***:** The figures are moved to the appropriate positions in the main content. Thanks.

*Summary and recommendation*

*In my opinion the paper needs substantial corrections to meet the standards of the HESS.*

*The novelty of the applied methodology is dubious and ther results obtained are unreliable. I would suggest also to re-phrase some parts of the article, because it is easy to lose the thread in the number of parameters, models, and methods applied in the study. Besides, the results should be generalised for other similar areas, otherwise they are only of local interest.*

*Response***:** The novelty of the paper is to explore the complex relationship between spatiotemporal rainfall structure and quick flood responses in small urbanized watersheds. The framework that combining high-resolution radar rainfall, stochastics storm transposition and GSSHA model provides an attractive approach for detailed flood frequency analysis. It can explore many questions that cannot be achieved by conventional frequency analysis which adopts several idealized assumptions. Without the approach, it is of difficulties to examine the rainfall-flood relationships under a statistical framework. We have rephrased the Introduction and discussed the limitation of the study in the last chapter. "Coupling the GSSHA model and SST-based rainfall frequency analysis, this study provides an effective

approach for regional flood frequency analysis for urban watersheds. Some idealized assumption used in the conventional method is questioned. It can be used to explore the dominant control on the upper tail of urban flood peaks, without many of the limiting assumptions associated with design storm methods. The study area could be extended in future work with larger basin scales and by manipulating the spatial heterogeneity of basin characteristics within GSSHA or other similar modeling systems."

Reference

Groffman, P. M., D. J. Bain, L. E. Band, K. T. Belt, G. S. Brush, J. M. Grove, R. V. Pouyat, I. C. Yesilonis, and W. C. Zipperer (2003), Down by the riverside: urban riparian ecology, *Frontiers in Ecology and the Environment*, 1(6), 315-321.

Lindner, G. A., and A. J. Miller (2012), Numerical Modeling of Stage-Discharge Relationships in Urban Streams, *Journal of Hydrologic Engineering*, 17(4), 590-596. doi:10.1061/(ASCE)HE.1943-5584.0000459

Seo, B.-C., W. F. Krajewski, A. Kruger, P. Domaszczynski, J. A. Smith, and M. Steiner (2011), Radar-rainfall estimation algorithms of Hydro-NEXRAD, *Journal of Hydroinformatics*, 13(2), 277-291. doi:10.2166/hydro.2010.003

Smith, B., J. Smith, M. Baeck, and A. Miller (2015), Exploring storage and runoff generation processes for urban flooding through a physically based watershed model, *Water Resources Research*, 51(3), 1552-1569. doi:10.1002/2014WR016085

Smith, B. K., J. Smith, and M. L. Baeck (2016), Flash flood-producing storm properties in a small urban watershed, *Journal of Hydrometeorology*, 7(2016), 2631–2647.

Smith, B. K., J. A. Smith, M. L. Baeck, G. Villarini, and D. B. Wright (2013), Spectrum of storm event hydrologic response in urban watersheds, *Water Resources Research*, 49(5), 2649-2663. doi:10.1002/wrcr.20223

Smith, J. A., and W. F. Krajewski (1991), Estimation of the mean field bias of radar rainfall estimates, *Journal of Applied Meteorology*, 30(4), 397-412.

Smith, J. A., M. L. Baeck, K. L. Meierdiercks, P. A. Nelson, A. J. Miller, and E. J. Holland (2005), Field studies of the storm event hydrologic response in an urbanizing watershed, *Water Resources Research*, 41(10), W10413(10415). doi:10.1029/2004wr003712

ten Veldhuis, M. C., Z. Zhou, L. Yang, S. Liu, and J. Smith (2018), The role of storm scale, position and movement in controlling urban flood response, *Hydrology and Earth System Sciences*, 22(1), 417-436. 10.5194/hess-22-417-2018

Wright, D. B., J. A. Smith, and M. L. Baeck (2014a), Flood frequency analysis using radar rainfall fields and stochastic storm transposition, *Water Resources Research*, 50(2), 1592-1615. doi:10.1002/2013WR014224

Wright, D. B., R. Mantilla, and C. D. Peters-Lidard (2017), A remote sensing-based tool for assessing rainfall-driven hazards, *Environmental Modelling & Software*, 90, 34-54. doi:10.1016/j.envsoft.2016.12.006

Wright, D. B., J. A. Smith, G. Villarini, and M. L. Baeck (2012), Hydroclimatology of flash flooding in Atlanta, *Water Resources Research*, 48(4). https://doi.org/10.1029/2011wr011371

Wright, D. B., J. A. Smith, G. Villarini, and M. L. Baeck (2013), Estimating the frequency of extreme rainfall using weather radar and stochastic storm transposition, *Journal of Hydrology*, 488, 150-165. doi:10.1016/j.jhydrol.2013.03.003

Wright, D. B., J. A. Smith, G. Villarini, and M. L. Baeck (2014b), Long-term high-resolution radar rainfall fields for urban hydrology, *Journal of the American Water Resources Association*, 50(3), 713-734. doi:10.1111/jawr.12139

Yang, Y., L. Sun, R. Li, J. Yin, and D. Yu (2020), Linking a Storm Water Management Model to a Novel Two-Dimensional Model for Urban Pluvial Flood Modeling, *International Journal of Disaster Risk Science*, 11(4), 508-518. 10.1007/s13753-020-00278-7

Yin, J., D. Yu, Z. Yin, M. Liu, and Q. He (2016), Evaluating the impact and risk of pluvial flash flood on intra-urban road network: A case study in the city center of Shanghai, China, *Journal of Hydrology*, 537, 138-145. https://doi.org/10.1016/j.jhydrol.2016.03.037

Yu, G., D. B. Wright, Z. Zhu, C. Smith, and K. D. Holman (2019), Process-based flood frequency analysis in an agricultural watershed exhibiting nonstationary flood seasonality, *Hydrol. Earth Syst. Sci.*, 23(5), 2225-2243. doi:10.5194/hess-23-2225-2019

Zhou, Z., J. A. Smith, D. B. Wright, M. L. Baeck, and S. Liu (2019), Storm catalog-based analysis of rainfall heterogeneity and frequency in a complex terrain, *Water Resources Research*, 55(3), 1871-1889. doi:10.1029/2018WR023567

Zhuang, Q., S. Liu, and Z. Zhou (2020), Spatial Heterogeneity Analysis of Short-Duration Extreme Rainfall Events in Megacities in China, *Water*, 12(12), 3364.

---

## Author Comment (AC2)

**Response to Anonymous Referee #2**

Comments/text of reviewer posted in **bold** *and italics*; the authors's answeres start with "***Response***:"; the sentences in the revised version is in blue.

Anonymous Referee #2

*Referee comment on "The impact of spatiotemporal structure of rainfall on flood frequency over a small urban watershed: an approach coupling stochastic storm transposition and hydrologic modeling" by Zhengzheng Zhou et al., Hydrol. Earth Syst. Sci. Discuss.,*
*https://doi.org/10.5194/hess-2021-161-RC2, 2021*

*This paper addresses the issue of the impact of small scale spatio-temporal rainfall variability on the hydrologic response of small urbanized catchment. The topic is relevant for the community and HESS journal. The paper is well written, and rather straightforward to read. However I believe that major modifications are needed before a potential publication in HESS. I have three major comments :*
*- Results are often not so clear with contrasting trends. In order to clarify the outcome of the study, I think that statistical significance of results should be quantified more systematically. This could help to clarify some results.*

***Response***: Thank you for your suggestions. The significance level of rainfall frequency is done in the study of Zhou *et al.* (2019). In this study, we use the results of rainfall frequency and then simulate the flood discharges based on the rainfall results. The upper and lower level of flood simulation can be found in Figure 5 in the revised version.

[Figure]

**Figure 5. Time series of simulated hydrographs for Franklintown based on the 3-h design storms from 10-yr to 200-yr return periods with spatially uniform (blue) and spatially distributed (red) rainfall. The grey bar indicates the median value of basin-averaged rainfall rate.**

*- Physical interpretation of the results should also be systematically provided, with quantification of the various effect implemented (notably the percentage of imperviousness of the various catchments). This would enable to enlarge the potential application of the paper's results to other locations.*

*__Response__*: Generally, the differences in hydrologic responses among urban basins cannot be fully accounted for by differences in impervious cover (Zhou *et al.*, 2017). For example, we compared the flood peaks at DR1, DR2 and DR5 which have similar basin scales (1.32, 1.92 and 2.05 km$^2$, respectively). DR1 with higher imperviousness and detention controlled area than DR2, has higher flood peaks than DR2. However, DR5 with lower imperviousness and detention controlled area has lower peaks than DR1 and DR2. A common agreement is that imperviousness tends to increase flood peaks while detention infrastructure tends to decrease peaks. Thus, the comparison among the three small-scale basins show that difficulties remain in attributing specific changes in urban flood peak distributions to specific urbanization characteristics. We have addressed the complexities in urban flood responses among the DR watershed, which can provide implementations for other urban watersheds. The role of urban features in flood responses is beyond the scope of this paper. Thank you.

*- It is not clear at what spatial scale are the rainfall return periods estimated ? This could have a strong impact on the results.*

*__Response__:* The rainfall frequency is estimated for the entire DR watershed with a spatial scale of 14.3 km$^2$.

*Detailed comments :*

*2) Data and methods*

*2.1) Study region and data*

*- Fig. 1.b : could you please clarify the configuration of the catchment, and notably the path of the various rivers through the catchment, which do not seem to converge toward the outlet of the catchment ? Please also update the colours because the map is difficult to read.*

*__Response__:* Figure 1 has been modified. The outline of streams are clarified in Fig.1b. The colors are changed to present the main types of land use land cover. For the details of the configuration of watershed used in GSSHA model, the readers are directed to Smith *et al.* (2015).

[Figure]

**Figure 1. Overview of Dead Run study region including (a) location of DR, elevation, and transposition domain of SST; (b) land use land cover and stream gages. The red outline and grey outline in (a) indicates the boundary of DR watershed and Baltimore City, respectively.**

*- Table 1 : please clarify what is "developed land" and "controlled area".*

***Response****:* The note is added in the end of Table 1 in the revised version as follows.

"Note:

a. Developed lands include "Developed, open space" (>20% impervious surface), "Developed, low intensity" (20%-49% impervious surface), "Developed, medium intensity" (50%-79% impervious surface), and "Developed, high intensity" (80% or more impervious surface). Data source: USGS 2012 National Land Cover Dataset (NLCD).

b. Dentention controlled areab refers to the area controlled by detention infrastructure."

- l. 94-98 : sentence weirdly written, please rephrase

***Response****:* We have rephrased the paragraph and added a general introduction of bias correction for radar data. In the revised version, it is written as follows. "High resolution (15-min temporal resolution, 1-km$^2$ spatial resolution) 34radar rainfall fields for the 2000-2015 period were derived from volume scan reflectivity fields from the Sterling, Virginia WSR-88D (Weather Surveillance Radar-1988 Doppler) radar. The two-dimentional radar rainfall fields are then developed from the reflectivity fields using the Hydro-NEXRAD algorithms (Krajewski *et al.*, 2011) which have been used in rainfall and hydrological studies (Smith *et al.*, 2007; Lin *et al.*, 2010; Smith *et al.*, 2013; Wright *et al.*, 2014; Zhou *et al.*, 2017). The Hydro-NEXRAD algorithms includes quality control algorithms, *Z-R* conversion of reflectivity to rainfall rate, time integration, and spatial mapping algorithms (Seo *et al.*, 2011). To improve the rainfall estimates, a multiplicative mean-field bias correction (Smith and Krajewski, 1991; Wright *et al.*, 2012) is applied on a daily basis using a network of 54 rain gauges in and around the Baltimore County. The bias computation takes the form $B_i = \frac{\sum_{S_i} G_{ij}}{\sum_{S_i} R_{ij}}$. Where $G_{ij}$ is the rainfall accumulation for gage $j$ on day $i$, $R_{ij}$ is the daily rainfall accumulation for the co-located radar pixel accumulation on day $i$, and $S_i$ is the index of the rain gage stations for which both the rain gage and the radar report positive rainfall accumulations for day $i$. Each 15-min radar rainfall field from day $i$ is then multiplied by $B_i$."

*2.2) GSSHA hydrological model*
*- Please explain how the interactions between surface flow and stormwater system is handled, because it is crucial in highly impervious areas.*

***Response****:* The 2-D overland flow grid empties into 1-D stream channels represented by cross sections that contain both the stream channel and the floodplain. Surface water is accumulated until the specified retention depth of the cell is exceeded. The overland flow is then routed in two orthogonal directions using Manning's equation with the diffusion wave form of the de St. Venant equations to estimate friction slope. When the overland flow reaches a model grid cell that contains a channel node, the flow is passed into the channel and routed using a 1-D technique through the stream network.

Culverts were measured in the field and represented in-stream as cross sections. Storm pipes were represented through cross sections with a rounded half-circle bottom and walls reaching up vertically. The circle's diameter was set to the pipe diameter. Detention basins were represented within the channel with cross sections extracted from the 1-m lidar topographic data. Water was backed up within the detention basins using rating curves at the detention basin outlets. The rating curves were based on a simple orifice equation, calculated for a fully submerged outlet pipe with a size determined by the downstream pipe size. Over all streams, culverts, pipes, and detention basins 989 cross sections were included within the model for an average spacing of 72.8 m between distinct cross sections.

*2.3) SST procedure*

*- Over which area is rainfall computed to estimate the return periods ?*

*Response**:* The rainfall is estimated for the entire DR watershed (14.3 km$^2$) with spatiotemporal structure.

*- l. 153 : "to generate estimates for return periods up to 500 years". I believe an assessment of the corresponding uncertainties should be provided.*

*Response:* In the study of *Zhou et al.* 2019, 1000 realizations of 500-yr series are generated, and the median values of 1000 realizations are used to generate estimates for return periods up to 500 years. The 90$^{th}$ and 10$^{th}$ quantiles of 1000 realizations are set as the lower and upper bound of SST estimates (Figure R1). The uncertainties of rainfall estimations are demonstrated in detail in *Zhou et al.* 2019.

[Figure]

Figure R1. The SST- based intensity-duration-frequency curves.

*2.4) Characteristics of rainfall and hydrologic response*

*- Eq 1 : given it is a basin average at time t, I guess that the integral should not be over time (T), but over the spatial domain.*

*Response*: Eq.1 has been revised as $\mathrm{M(t)} = \int_A R(t,x)dx$. Thanks.

*- Eq. 3 : if M(t) is a rain rate, I guess the time step should appear in the computation of the cumulative depth. More generally, please indicate units of all the quantities used.*

*Response*: The units of the quantities are labeled as "Peak basin-average rainfall rate (mm/h)" and "storm total rainfall depth (mm)". Thanks.

*- l. 182-183 : please clarify what you are calling a "unimodal distribution for rainfall".*

*Response*: The unimodal distribution indicates that there is spatially one peak over the entire watershed. It is clarified in the revised version.

*- Are all the indicators really used ? I do not have the feeling that they are really all used and that the high number is just creating a bit of confusion. I would either really exploit all of them or reduce their number.*

*Response*: The high-resolution radar rainfall data provides us a good opportunity to examine the details of rainfall structure. We then extract the indicators include temporal characteristics of rainfall (rainfall peak, rainfall rate and total rainfall) and spatial characteristics of rainfall (spatial coverage, rainfall location and rainfall-weighted flow distance). By using these indicators, the rainfall structure can be

described comprehensively. And then coupled with a hydrological model, we can capture the rainfall-flood relation well. Through the examination of all the indicators, the main features that impact flood responses can be found.

Therefore, we prefer to keep all the indicators in the content. They are all mentioned in the results section. A complimentary demonstration is shown in Figure A3, showing the relationship between spatiotemporal rainfall structure and flood responses. The indicators that are highly related to flood responses are highlighted and addressed.

[Figure]

Figure A3: Correlation between space-time rainfall structure and flood responses at Franklintown under 10-yr and 200-yr return periods.

**3) Result and discussion**

**3.1) Model validation**

- *I believe that an example of hydrographs should be included in the main document.*

*Response:* Thank you for the suggestion. The hydrograph of the 14 August 2011 storm event is added as a representative simulation (Figure 3 in the revised version).

[Figure]

**Figure 3. Hydrographs and rainfall for the the 14 August 2011 storm event. Time refers to minutes from the start of the model simulation.**

*- The differences observed in Fig. 2.a are quite significant and should be more discussed.*

**Response***:* Thank you for the suggestion. We have revised the paragraph to make it clearer. The flood magnitude at the downstream Franklintown gage is well captured with the median peak discharge difference of -14%. The largest median peak discharge difference of 57% is at DR1. The reason is likely that it has a large area of land which was not represented fully on county storm sewer maps (Smith *et al.*, 2015). A similar situation is found in Smith *et al.* (2015), and their research shows the model can be effective in producing hydrological responses. As mentioned in Section 2.1, the rating curves used to compute the discharge data at DR3 and DR4 are not provided from USGS. It may increase the error in the measurements and modeling results. Overall, for a large collection of flood events with various rainfall characteristics and peak discharges ranging from 70 $m^3$/s to 253 $m^3$/s, the model performs well to reproduce peak discharges, especially at the basin outlet.

The discussion of error from discharge is added in the revised version as follows. "It should be noted that the error in simulated response may be attributable to measurement errors tied to stage discharge curves and to conversions of radar reflectivity to rainfall rate, as well as to the features that were simplified within the model, such as initial soil moisture and some aspects of the storm drain network (Smith *et al.*, 2015). For example, it has been documented that the average error of discharge between USGS direct measurements and stage-discharge curves for Franklintown is 17.4% between 2008 and 2010 (Lindner and Miller, 2012). As mentioned in Section 2.1, the rating cruve used to compute the discharge data at DR3 and DR4 is not provided from USGS. It may increase the error in the measurements and modeling results. For the rainfall data set used in this study, the difference of the storm total rainfall between a rain gage and the bias-corrected radar rainfall data for the pixel of that gage is compared. The median difference for all gages over the 21 storms is 22.6%(Smith *et al.*, 2015)."

*- Also, limiting the validation of the model to peak discharge and time of peak do not seem sufficient. Adding indicators using the whole hydrographs (such as the imperfect Nash-Sutcliffe efficiency and not only the skewness) would add some relevancy to the validation.*

*Response:* Thank you for your suggestion. NSE is added in the revised version as follows. "The median Nash-Sutcliffe Efficiency (NSE) for the 21 events at Franklintwon is 0.77 (Fig. 2c). The best NSE at Franklintown is 0.97 indicating that the match between model and measured data was nearly exact. For the subwatersheds, the best median NSE is at DR-4 with a value of 0.74, while the least median NSE is at DR-1 with a value of 0.21. The results show that the main tendency of flood response is captured by the model."

[Figure]

**Figure 2. Comparison of (a) flood peak discharges, (b) response times and (c) NSE for 21 historical rainfall events.**

*- Fig 3.a : clarify how normalization is implemented. Also more explanation/interpretation of the multimodal behaviour for flood should be added. Is it due to location of rainfall ? Is the same behaviour also observed for the other stations.*

*Response:* The equation for normalization is added in the revised version as follows. "The normalization is the ratio of values minus the minimum to the maximum minus minimum." The multimodal behavior for flood is caused by the mixed impact of rainfall process and the urban drainage system. We cannot attribute the behavior simply to rainfall nor the urban drainage system. The following results show that flood peak is more correlated with spatial rainfall features, implying that the multimodal distribution of flood peaks is more likely associated with spatial rainfall distributions. We have discussed in the revised version as follows. "The following results will show that flood peak is more correlated with spatial rainfall features, implying that the multimodal distribution of flood peaks is more likely associated with the spatial distribution of rainfall."

*- l. 242-243 : please provide some explanations / interpretations to this retrieved behaviour.*

*Response:* The interpretation is added in the revised version as follows. "For the 100-yr return period, the upper spread shows a tendency toward dual peaks, which cannot be revealed from conventional design flood practices. Since in the conventional rainfall flood frequency approach, the design storm is temporally idealized as a unimodal peak process. By using theses design storm, the flood response is generally simulated as a unimodal peak process. The above results imply the uncertainty and insufficiency of flood frequency analysis in the conventional method. For the 200-yr return period, the hydrograph is peakiest with a large upper spread."

*- Fig. 5 and associated comments : the fact that no clear trend with basin size is found is somehow surprising. Differences in land use are mentioned and should be further explored and quantified I*

*think. Also, was a sensitivity analysis carried out on the choice of rainfall events ?*

*Response**:* Generally, basin scale plays an important role in determining the distribution of flood magnitudes. We discussed the difference in land use and linked the difference with the difference in flood peaks. However, the impacts are subtle, and the data are not robust enough to do more than suggest these implementations.

We did not manually select rainfall events to drive the hydrologic model. Based on the SST framework, the rainfall events are selected and transposed randomly. Thus, for each return period, the rainfall results are not from one or several specific rainfall events. A sensitivity analysis is not needed.

**3.3) Rainfall-Flood relationship**

*- l. 284 : a correlation of 0.16 is very low, so I am not sure the wording "somewhat correlated is appropriate"*

*Response**:* The sentence is revised as "the flood peak is slightly correlated with…"

*- Fig 7 and associated comments : please provide more insights about the "random forest regression model" for the non-specialist reader. Also, a lot of rainfall features while floods are quantified simply by flood peak. How significant are the differences between 10yr and 200 yr results ? Would a different selection of initial rainfall events lead to different results?*

*Response**:* The general introduction of RF model is discussed in the revised version "Random forests (RF) is an ensemble learning method (Breiman, 2001) that aggregates results from multiple models to achieve better accuracy. RF is one of the most widely-used method for regression and classification. Moreover, it is relatively easy to train and tests. In this study, rainfall spacetime structure characteristics are used as RF model features. The flood peak is set as the model target. The relationship between rainfall structure and flood peak is then explored under the RF-based regression method.".

The magnitude of flood peak is the main target in flood frequency analysis. Furthermore, in this study, the response time is less impacted in the small urbanized watershed. We hence focus on the flood peak. According to the values of feature importance, the importance of the total rainfall decreased by 2% compared with the 10-yr return period. The largest difference is for the fractional coverage of storm core which increases by 5% from 10-yr to 200-yr return period. Both rainfall total and rainfall peak decrease from small return periods to large return periods; on the contrary for the dispersion of RWD ($S$) and fractional coverage of storm core ($Z$). Though it appears that the difference is moderate, but for a such small watershed, the tendency of the change of spatiotemporal rainfall feature importance is noteworthy. We have mentioned it in the revised version as follows. "For extreme storms, the maximum discharge is more closely linked to the spatial structure of rainfall, which is consistent with the results in (Peleg *et al.*, 2017; Zhu *et al.*, 2018). Though it appears that the difference is moderate, but for a such small watershed, the tendency of the change of spatiotemperol rainfall feature importance is noteworthy."

It is possible that the different rainfall events will lead to different results. But unlike many previous studies which used a limited number of rainfall events, in this study, we selected a storm catalog with 200 events with various spatiotemporal structures. And their transposed results are then further computed as the rainfall frequency analysis. For each return period, the 300 transposed events contain various rainfall structures which can provide more general and reasonable results.

*- l. 306-309 : It is not obvious to me how the general conclusion is obtained from the Fig. 8. Please clarify ? Notably, the figure does not provide answers on the rapidness of flood response if I*

*understood it well …*

*Response:* The Figure 8 and the discussion are removed from the manuscript. Thanks.

*- l. 311-318 : I believe that the paragraph is actually quite interesting, and that more interpretation should be done. However, the spatial scale used to determine the rainfall return period should be clarified and is likely to play a significant role, notably in relation with the size of the studied catchment. For which of the catchments was Fig. 9 obtained ? Are different trends found for other catchments ?*

*Response:* The rainfall results are for the entire watershed (14.3 km$^2$) and the flood frequency results are also for the same watershed. The limitation of this analysis is that the total number of realizations is only 30 with return periods up to 100 years. With more run of the model, it is possible to obtain some functional relationship between flood return periods and rainfall return periods, which could be done in our future work.

As shown in the following figure, similar results are found for the sub-basins (flood return periods at sub-basins vs. rainfall return periods at the DR outlet).

[Figure]

**Figure R3. Scatterplot of return periods for DR-scale rainfall and subbasin-scale peak discharge.**

**Reference**

Breiman, L. (2001), Random Forests, *Machine Learning*, 45(1), 5-32. 10.1023/a:1010933404324

Krajewski, W. F., A. Kruger, J. A. Smith, R. Lawrence, C. Gunyon, R. Goska, B.-C. Seo, P.

Domaszczynski, M. L. Baeck, and M. K. Ramamurthy (2011), Towards better utilization of NEXRAD data in hydrology: an overview of Hydro-NEXRAD, *Journal of hydroinformatics*, 13(2), 255-266. doi:10.2166/hydro.2010.056

Lin, N., J. A. Smith, G. Villarini, T. P. Marchok, and M. L. Baeck (2010), Modeling extreme rainfall, winds, and surge from Hurricane Isabel (2003), *Weather and forecasting*, 25(5), 1342-1361. doi:10.1175/2010WAF2222349.1

Lindner, G. A., and A. J. Miller (2012), Numerical Modeling of Stage-Discharge Relationships in Urban Streams, *Journal of Hydrologic Engineering*, 17(4), 590-596. doi:10.1061/(ASCE)HE.1943-5584.0000459

Peleg, N., F. Blumensaat, P. Molnar, S. Fatichi, and P. Burlando (2017), Partitioning the impacts of spatial and climatological rainfall variability in urban drainage modeling, *Hydrology and Earth System Sciences*, 21(3), 1559. doi:10.5194/hess-21-1559-2017

Seo, B.-C., W. F. Krajewski, A. Kruger, P. Domaszczynski, J. A. Smith, and M. Steiner (2011), Radar-rainfall estimation algorithms of Hydro-NEXRAD, *Journal of Hydroinformatics*, 13(2), 277-291. doi:10.2166/hydro.2010.003

Smith, B., J. Smith, M. Baeck, and A. Miller (2015), Exploring storage and runoff generation processes for urban flooding through a physically based watershed model, *Water Resources Research*, 51(3), 1552-1569. doi:10.1002/2014WR016085

Smith, B. K., J. A. Smith, M. L. Baeck, G. Villarini, and D. B. Wright (2013), Spectrum of storm event hydrologic response in urban watersheds, *Water Resources Research*, 49(5), 2649-2663. doi:10.1002/wrcr.20223

Smith, J. A., and W. F. Krajewski (1991), Estimation of the mean field bias of radar rainfall estimates, *Journal of Applied Meteorology*, 30(4), 397-412.

Smith, J. A., M. L. Baeck, K. L. Meierdiercks, A. J. Miller, and W. F. Krajewski (2007), Radar rainfall estimation for flash flood forecasting in small urban watersheds, *Advances in Water Resources*, 30(10), 2087-2097. doi:10.1016/j.advwatres.2006.09.007

Wright, D. B., J. A. Smith, G. Villarini, and M. L. Baeck (2012), Hydroclimatology of flash flooding in Atlanta, *Water Resources Research*, 48(4). https://doi.org/10.1029/2011wr011371

Wright, D. B., J. A. Smith, G. Villarini, and M. L. Baeck (2014), Long-term high-resolution radar rainfall fields for urban hydrology, *Journal of the American Water Resources Association*, 50(3), 713-734. doi:10.1111/jawr.12139

Zhou, Z., J. A. Smith, D. B. Wright, M. L. Baeck, and S. Liu (2019), Storm catalog-based analysis of rainfall heterogeneity and frequency in a complex terrain, *Water Resources Research*, 55(3), 1871-1889. doi:10.1029/2018WR023567

Zhou, Z., J. A. Smith, L. Yang, M. L. Baeck, M. Chaney, M.-C. Ten Veldhuis, H. Deng, and S. Liu (2017), The complexities of urban flood response: Flood frequency analyses for the Charlotte Metropolitan Region, *Water Resources Research*, 53(8), 7401-7425. doi:10.1002/2016WR019997

Zhu, Z., D. B. Wright, and G. Yu (2018), The Impact of Rainfall Space-Time Structure in Flood Frequency Analysis, *Water Resources Research*, 54(11), 8983-8998. doi:10.1029/2018wr023550